# Multigrid-Augmented Deep Learning Preconditioners for the Helmholtz Equation using Compact Implicit Layers

**Ido Ben-Yair** *
Department of Computer Science
Ben-Gurion University of the Negev
idobeny@post.bgu.ac.il

**Bar Lerer** *
Department of Computer Science
Ben-Gurion University of the Negev
barlere@post.bgu.ac.il

**Eran Treister**
Department of Computer Science
Ben-Gurion University of the Negev
erant@cs.bgu.ac.il

## Abstract

We present a deep learning-based iterative approach to solve the discrete heterogeneous Helmholtz equation for high wavenumbers. Combining classical iterative multigrid solvers and neural networks via preconditioning, we obtain a faster, learned neural solver that scales better than a standard multigrid solver. We construct a multilevel U-Net-like encoder-solver CNN with an implicit layer on the coarsest level, where convolution kernels are inverted. This alleviates the field of view problem in CNNs and allows better scalability. Furthermore, we propose a multiscale training approach that enables to scale to problems of previously unseen dimensions while still maintaining a reasonable training procedure.

## 1 Introduction

The Helmholtz equation is a partial differential equation (PDE) that models the propagation of waves in the frequency domain. This equation occurs in many disciplines of engineering and science. However, in complex real-world environments, an analytical solution is difficult to obtain. Thus, numerical methods are typically used, whether based on finite difference discretizations, iterative solvers, or many other approaches. Indeed, solving the discrete Helmholtz equation efficiently is a substantial field of research (Dwarka & Vuik, 2020; Gander & Zhang, 2019; Graham et al., 2020; Luo et al., 2014; Olson & Schroder, 2010; Poulson et al., 2013; Reps & Weinzierl, 2017; Sheikh et al., 2016; Treister & Haber, 2019).

One common method for the Helmholtz equation is multigrid. Such methods aim to complement standard local methods called relaxations, which attenuate only high-frequency error components efficiently. However, multigrid alone does not perform well for the Helmholtz equation, mostly due to the indefiniteness of the resulting linear system. Hence, the shifted Laplacian (SL) approach is often used (Erlangga YA, 2006; Umetani et al., 2009; Elman et al., 2001; Erlangga et al., 2004), where the Helmholtz operator is shifted by an imaginary term. SL works well when used, for example, as a preconditioner to a Krylov method.

Neural networks are known as universal approximators, i.e., capable of representing any smooth signal. However, it has been established that to represent highly oscillatory functions, deep networks require either substantial depth or other special considerations (Rahaman et al., 2019; Tancik et al., 2020; Li et al., 2021; Sitzmann et al., 2020). Thus, we expect deep networks to scale poorly with the complexity of the problem. Successful uses of deep learning for solving highly oscillatory PDEs must therefore find ways to deal with the high frequencies inherent to this type of data.

---

*Ido Ben-Yair and Bar Lerer have contributed equally to this research.

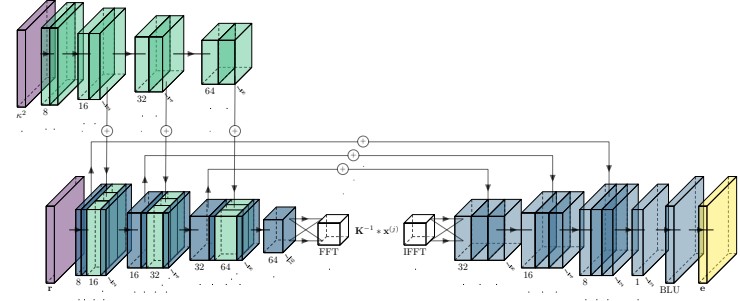

Figure 1: Implicit encoder-solver CNN architecture. *(Top)*The encoder computes feature maps which are added to the solver as indicated by the arrows. *(Bottom)*The solver maps a residual vector **r** to an error **e**. BLU stands for bilinear upsampling.

To achieve good performance during training, and enable successful generalization, here we exploit the close connection between convolutional neural networks and multigrid cycles: A U-Net (Ronneberger et al., 2015) is trained to act as a preconditioner to FGMRES (Saad, 1993). Thus, **our main contribution** is the introduction of an implicit layer at the coarsest level of the U-Net to mimic an exact coarse grid solution. In addition, we propose a multiscale training method where the training alternates between smaller and larger, hence more difficult problems. This enables the network to learn salient features quickly in smaller domains, reducing overall training time. The networks are shown to generalize to larger unseen sizes after being exposed to a smaller number of larger-scale problems.

## 2 THE HELMHOLTZ EQUATION

The heterogeneous Helmholtz equation is given by

$$-\Delta u(\vec{x}) - \omega^2 \kappa(\vec{x})^2 (1 - \gamma i) u(\vec{x}) = g(\vec{x}), \quad \vec{x} \in \Omega. \tag{1}$$

The unknown $u(\vec{x})$ is the Fourier-space representation of the pressure wave function, while $\omega$ denotes the angular frequency of the wave, $g(\vec{x})$ represents any sources present, $\Delta$ is the Laplacian operator, $i = \sqrt{-1}$, and $\kappa(\vec{x})$ denotes the heterogeneous wave slowness model. $\gamma$ indicates the fraction of global damping in the medium, which is assumed to be very small and constant.

Equation (1) is then discretized using second-order finite-differences on a uniform 2D grid of width $h$ in both dimensions, which yields a global linear system,

$$A^h \mathbf{u}^h = \mathbf{g}^h, \tag{2}$$

where $A^h$ is the operator matrix:

$$A^h = -\Delta_h - \omega^2 \kappa(\mathbf{x})^2 I = \frac{1}{h^2} \begin{bmatrix} 0 & -1 & 0 \\ -1 & 4 - \omega^2 \kappa(\mathbf{x})^2 h^2 & -1 \\ 0 & -1 & 0 \end{bmatrix}. \tag{3}$$

However, since standard multigrid methods struggle to deal with the indefiniteness of eq. (3), we use the shifted Laplacian operator

$$-\Delta u - \omega^2 \kappa(\vec{x})^2 (\alpha - \beta i) u, \quad \alpha, \beta \in \mathbb{R}, \tag{4}$$

instead as a preconditioner in FGMRES. The SL multigrid method is consistent and robust for heterogeneous slowness models, i.e., where $\kappa$ is not uniform. However, it is considered slow and computationally expensive, especially for high wavenumbers, which is our interest.

## 3 A MULTIGRID-AUGMENTED CNN PRECONDITIONER

Our method is based on the U-Net encoder-solver method, proposed by Azulay & Treister (2021). This U-Net architecture is similar in spirit to the multigrid V-cycle and shares common properties.

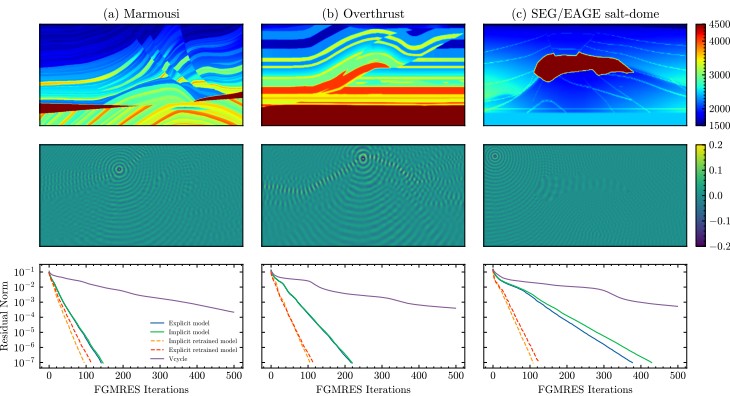

Figure 2: Out-of-distribution test. *(Top)* Velocity models used for each test. *(Middle)* The solution to a single-source Helmholtz problem computed by a single application of FGMRES with the implicit network, followed by a V-cycle, as preconditioner. *(Bottom)* Convergence plots of the implicit and explicit network preconditioners on each respective problem, as well as a V-cycle-only preconditioner.

However, it suffers from a field of view problem as it has a limited number of levels. Hence, we propose a novel implicit layer inspired by the Lippmann-Schwinger (LiS) equation. This encoder-solver implicit network is shown in fig. 1. To solve many instances of the Helmholtz equation, we also train an encoder to be applied before the solver network.

**The encoder network** emits a latent representation of the heterogeneous parameters in $\kappa^2$, while the solver network is trained to use these encodings. **The solver network** maps the residuals to corresponding error vectors, which are then fed into another, non-learned, V-cycle for further smoothing before being passed back to FGMRES. Hence, we train the network on residual and error vectors that are not treated well by V-cycles, and allow the network to generate noisy predictions because these can be attenuated well by the V-cycle. This way we simplify the learning task of the CNN, and show that this combination significantly reduces the number of FGMRES iterations required.

To obtain a solution, the slowness model is first encoded; then, to smooth out the initial error, we start with a few iterations of FGMRES with the V-cycle only as a preconditioner. The result is used again in FGMRES, where now the solver network, followed by a V-cycle, act as a preconditioner. Since the encodings are computed once per slowness model and remain fixed, their cost is amortized over many invocations of the solver network. This is especially desirable when solving many times for the same $\kappa^2$, such as when solving inverse problems. Both the encoder and solver networks are trained together to ensure compatibility.

As the U-Net architecture suffers from a field of view problem, caused by our training on small grids, we propose a novel layer that mimics the exact solution of the coarsest grid in V-cycles, which has a *global* field of view. Here, we build on the approach of Haber et al. (2019) which inverts a compact convolution kernel. However, their formulation did not work well in our setup, likely due to sensitivity to the boundary conditions, as the FFT considers periodic BCs, while we use absorbing BCs. Thus, to make our implicit layer suitable for Helmholtz matrix inversion, we take inspiration from the LiS equation, which has been used to solve inverse problems that feature the Helmholtz equation (Soubies et al., 2017; Pham et al., 2020):

$$\mathbf{u}^h + A^h(\kappa_0)^{-1}\omega^2(\kappa_0^2 - \kappa^2(\mathbf{x}))\mathbf{u}^h = A^h(\kappa_0)^{-1}\mathbf{g}^h, \tag{5}$$

where $A^h(\kappa_0)^{-1}$ is the inverse of a Helmholtz operator with a constant medium $\kappa_0$, which can be modeled by a fixed kernel. This is equivalent to a convolution with a suitable Green's function sampled on a twice larger grid at each dimension. This Green's function is then convolved at each application of the matrix in using FFT.

Let $\widehat{\mathbf{G}}_K$ be the Fourier transform of a Green's function $\mathbf{G}_K$ of a kernel $K$ twice the size at each dimension. The implicit layer is given by:

$$\mathbf{x}^{(j+1)} = \mathcal{F}^{-1}(\widehat{\mathbf{G}}_K \odot \mathcal{F}(\mathbf{x}^{(j)})), \tag{6}$$

Table 1: Comparison of preconditioner methods. Each type of network was trained on problems up to the grid size indicated in the columns to the right. The average number of FGMRES iterations are shown for a slowness model unseen during training and 1000 right-hand sides. The methods are executed until the residual norm falls by a factor of $10^{-7}$ or 2000 iterations are performed.

| | Preconditioner \ Grid size: | $128^2$ | $256^2$ | $512^2$ | $1K^2$ | $2K^2$ | $4K^2$ |
|---|---|---|---|---|---|---|---|
| **OpenFWI** | Shifted Laplacian V-cycle | 112.9 | 246.8 | 586.9 | 1983.8 | $> 2000$ | $> 2000$ |
| | Explicit U-Net (up to $128^2$) | 18.38 | 30.51 | 52.93 | 178.15 | $404.05^\dagger$ | $> 2000$ |
| | Implicit U-Net (up to $128^2$) | 17.26 | 28.98 | 50.63 | 126.09 | $502.34^\dagger$ | $> 2000$ |
| | Explicit U-Net (up to $256^2$) | 18.55 | 30.42 | 52.97 | 184.95 | $332.02^\dagger$ | $> 2000$ |
| | Implicit U-Net (up to $256^2$) | 17.21 | 28.52 | 47.90 | 92.31 | $281.20^\dagger$ | $> 2000$ |
| | Explicit U-Net (up to $512^2$) | 20.53 | 33.29 | 55.31 | 102.41 | 135.74 | 202.96 |
| | Implicit U-Net (up to $512^2$) | 18.82 | 27.34 | 40.62 | 63.39 | 94.13 | 143.03 |
| **STL-10** | Shifted Laplacian V-cycle | 166.2 | 360.72 | 750.93 | 1876.1 | $> 2000$ | $> 2000$ |
| | Explicit U-Net (up to $128^2$) | 25.85 | 44.66 | 75.96 | 233.60 | 284.30 | $> 2000$ |
| | Implicit U-Net (up to $128^2$) | 25.46 | 45.19 | 80.67 | 222.85 | $> 2000$ | $> 2000$ |
| | Explicit U-Net (up to $256^2$) | 28.11 | 34.60 | 54.44 | 77.75 | 139.12 | 280.43 |
| | Implicit U-Net (up to $256^2$) | 27.13 | 33.21 | 48.26 | 69.21 | 132.66 | 232.12 |
| | Explicit U-Net (up to $512^2$) | 27.42 | 34.42 | 54.67 | 77.52 | 155.29 | 231.72 |
| | Implicit U-Net (up to $512^2$) | 26.13 | 33.77 | 47.54 | 63.50 | 130.43 | 189.67 |
| **CIFAR-10** | Shifted Laplacian V-cycle | 97.3 | 245.4 | 545.9 | 1964.07 | $> 2000$ | $> 2000$ |
| | Azulay & Treister (2021) | 25 | 52 | 101 | N/A | N/A | N/A |
| | Explicit U-Net (up to $128^2$) | 19.10 | 30.61 | 49.64 | 88.19 | 172.02 | 224.51 |
| | Implicit U-Net (up to $128^2$) | 18.12 | 28.35 | 43.89 | 70.84 | 120.12 | 200.41 |
| | Explicit U-Net (up to $256^2$) | 18.25 | 28.96 | 49.47 | 84.40 | 167.18 | 216.75 |
| | Implicit U-Net (up to $256^2$) | 17.21 | 26.91 | 44.98 | 72.43 | 119.95 | 198.16 |
| | Explicit U-Net (up to $512^2$) | 16.76 | 28.73 | 44.51 | 75.32 | 108.33 | 171.15 |
| | Implicit U-Net (up to $512^2$) | 16.75 | 28.03 | 43.29 | 68.08 | 85.20 | 117.29 |

where $\mathcal{F}$ and $\mathcal{F}^{-1}$ denote the Fourier transform and its inverse, applied per-channel, and $\odot$ denotes the elementwise product. We train the implicit layer to learn $K$. To obtain the Green's function of some kernel $K$ in Fourier space needed for eq. (6), we divide the kernel's Fourier transform against a point source centered in the domain, as this division corresponds to matrix inversion using a fixed kernel:

$$\mathbf{G}_K = \mathcal{F}^{-1} \left( \frac{\mathcal{F}(K_{\mathrm{pad}})}{\mathcal{F}(K_{\mathrm{pad}})^2 + \epsilon} \odot \mathcal{F}(\delta) \right), \tag{7}$$

where $\epsilon = 10^{-5}$. To compute $\mathbf{G}_K$ using the inverse FFT in eq. (7) while reducing the influence from the boundaries, we zero-pad the domain to three times the size, and crop it afterward, that is: $\widehat{\mathbf{G}}_K = \mathcal{F}(\mathrm{crop}(\mathbf{G}_K))$. We note that the Green's functions in Fourier space $\widehat{\mathbf{G}}_K$ are computed only once for each kernel on the coarsest grid, as the kernels' weights do not depend on the input data. Furthermore, $\widehat{\mathbf{G}}_K$ are stored as part of the network and are not computed at all during inference. Lastly, since $\delta$ is a fixed point source, its FFT $\mathcal{F}(\delta)$ is fixed as well is computed only once per grid.

## 4 NUMERICAL RESULTS

To demonstrate the efficiency of the proposed techniques, we compare three preconditioning methods: an encoder-solver with an implicit U-Net followed by a shifted Laplacian V-cycle, an encoder-solver U-Net without the implicit step (called the *explicit U-Net*) also followed by a V-cycle, and lastly a V-cycle-only preconditioner. The results of network training are presented in table 1, which also shows that unless trained on the larger models of $512^2$, the networks may perform poorly or even fail to converge in larger test problems (e.g., $4K^2$). We also demonstrate out-of-distribution performance, where the distribution of the slowness model differs substantially from that seen during training. Figure 2 shows how the implicit and explicit networks, trained in earlier experiments, perform on these out-of-distribution models. The slowness models are generated from the CIFAR-10 (Krizhevsky, 2009), OpenFWI Style-A (Deng et al., 2022) and STL-10 (Coates et al., 2011) datasets. Details regarding the generation of data and multiscale training can be found in appendix D.

For more information about the training and architecture, see appendix F. Finally, additional results are given in appendix G.

## 5 Conclusion

In this work we introduced an implicit layer to the encoder-solver U-Net architecture, achieving faster convergence and overcoming a limited field of view. Additionally, we adopted architectural enhancements to speed up both the forward and backward computations of the CNNs.

## 6 Acknowledgements

This research was supported by the Israel Science Foundation grant no. 1589/19. IBY is supported by the Kreitman High-Tech scholarship at BGU.

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

## A ON SOLVING THE HELMHOLTZ EQUATION

The Helmholtz equation occurs in many different fields, where its real-world applications include seismic mappings of the earth, magnetic resonance imaging, optical diffraction tomography and more (Bernard et al., 2017; Soubies et al., 2017; Guasch et al., 2020; Pham et al., 2020). It is challenging to solve because it requires finer discretizations as the wavenumber $\omega\kappa$ grows. For large values of $\kappa$, the resulting linear system is highly indefinite (Bayliss et al., 1985; Haber & MacLachlan, 2011).

Typically, at least 10 grid nodes per wavelength are used in a discretization of the Helmholtz equation to obtain an accurate solution. This means $\omega\kappa h$ is typically bounded by $\omega\kappa h \leq \frac{2\pi}{10} \approx 0.628$. As a result, the required mesh can be very fine for high wavenumbers, which means that we require significantly more degrees of freedom. Hence, the system eq. (2) may become prohibitively large, ill-conditioned, and indefinite. Due to the boundary conditions, it is also complex-valued. Moreover, the eigenvalues of $A^h$ in eq. (3) will have more negative real parts as the wavenumber $\kappa\omega$ grows. Thus, solving large-scale systems of this kind often requires the use of efficient iterative solution techniques, like Krylov, multigrid, and other methods.

In addition, to model open domains, we use an absorbing boundary layer (ABL) (Engquist & Majda, 1977; 1979; Erlangga YA, 2006), i.e., a function in $\gamma$ that goes from 0 to 1 towards the boundaries. Sommerfeld, PML (Berenger, 1994; Singer & Turkel, 2004) or (Papadimitropoulos & Givoli, 2021) can be viable options as well.

## B ON THE CONNECTION BETWEEN CNNS AND LIPPMANN-SCHWINGER SOLVERS

A popular type of CNN is the ResNet architecture (He et al., 2016), which employs blocks of layers given by

$$\mathbf{x}^{(j+1)} = \mathbf{x}^{(j)} + \mathbf{K}_1^{(j)}\sigma\left(\mathbf{K}_2^{(j)}\mathbf{x}\right), \quad j = 0, \ldots, N-1, \tag{8}$$

where $\mathbf{x}^{(j)}$ and $\mathbf{x}^{(j+1)}$ are the input and output features respectively, $\mathbf{K}_1^{(j)}, \mathbf{K}_2^{(j)}$ are two different convolution operators, and $\sigma$ is the non-linear activation function.

To understand the connection between CNNs and implicit solvers, consider eq. (8) as a forward-Euler discretization of an underlying continuous non-linear ODE or PDE (Ruthotto & Haber, 2020),

$$\partial_t\mathbf{x}(t) = f(\mathbf{x}(t), \boldsymbol{\theta}(t)), \quad t \in [0, T], \tag{9}$$

where $f(\mathbf{x}(t), \boldsymbol{\theta}(t))$ is some non-linear function parameterized by $\boldsymbol{\theta}(t)$ and $[0, T]$ is a time interval which is discretized as per the number of layers in the network. The approach of Haber et al. (2019) suggested discretizing the time derivative in eq. (9) using an implicit (backward) Euler method instead of the explicit (forward) Euler, as in eq. (8). This implicit step is known to be effective in increasing the field of view, but it also requires an inversion of a convolution kernel rather than a multiplication, obtained using the Fast Fourier Transform (FFT) (Haber et al., 2019). This is appealing since here essentially we need to invert a spatially dependent kernel in eq. (3). Hence, a natural choice for the CNN solver would be an inverted convolution operator (i.e., implicit), albeit with a fixed (yet learned) kernel. We perform this step only at the coarsest level for several channels ($4\times$ coarsest in each dimension), hence we avoid the high cost associated with running the FFT on the fine feature maps.

To obtain the formulation of the implicit Lippmann-Schwinger layer, let us first consider the approach of Soubies et al. (2017), that with Haber et al. (2019) inspired the design of our implicit

layer. We note that the approach of Soubies et al. (2017) is applied in continuous space and then discretized, while here we present the idea on the discrete space. As motivation, consider eq. (3), where we add and subtract the term $\omega^2 \kappa_0^2 I$ with a constant $\kappa_0$:

$$A^h(\kappa(\mathbf{x})) = -\Delta_h - \omega^2 \kappa^2(\mathbf{x})I + \omega^2 \kappa_0^2 I - \omega^2 \kappa_0^2 I. \tag{10}$$

Hence, eq. (2) becomes

$$A^h(\kappa_0)\mathbf{u}^h + \omega^2(\kappa_0^2 - \kappa^2(\mathbf{x}))\mathbf{u}^h = \mathbf{g}^h. \tag{11}$$

The approach of Soubies et al. (2017) multiplies eq. (11) by $A^h(\kappa_0)^{-1}$ to form[1]

$$\mathbf{u}^h + A^h(\kappa_0)^{-1}\omega^2(\kappa_0^2 - \kappa^2(\mathbf{x}))\mathbf{u}^h = A^h(\kappa_0)^{-1}\mathbf{g}^h, \tag{12}$$

If $\kappa_0 \approx \kappa(x)$, then we have a well-conditioned system for $\mathbf{u}^h$ dominated by an identity matrix which is easy to solve. where $A^h(\kappa_0)^{-1}$ is the inverse of a Helmholtz operator with a constant medium $\kappa_0$ that can be modeled by a fixed kernel. This is equivalent to a convolution with a suitable Green's function sampled on a twice larger grid at each dimension. The Green's function is then convolved at each application of the matrix in using FFT. Because the Green's function is sampled on a large grid, there are no reflections or periodic continuations from the boundaries. In Soubies et al. (2017) the Green's function is defined analytically, and sampled.

We view the process in eqs. (11) and (12) as a preconditioned Helmholtz equation, where the preconditioner is the same operator but with constant media. That is, given an approximate solution $\mathbf{u}^h$ we approximate the error $\mathbf{e}^h$ by

$$\mathbf{e}^h \approx A^h(\kappa_0)^{-1}(\mathbf{g}^h - A^h(\kappa(\mathbf{x}))\mathbf{u}^h) = A^h(\kappa_0)^{-1}\mathbf{r}^h. \tag{13}$$

This preconditioner has a large field of view and can be implemented in a network efficiently on a GPU using FFT.

## C  GEOMETRIC MULTIGRID AND THE SHIFTED LAPLACIAN

A common multigrid method to solve the Helmholtz equation is the shifted Laplacian method, originally suggested by Erlangga YA (2006). Since standard multigrid methods struggle to deal with the indefiniteness of eq. (3), the shifted Laplacian operator

$$-\Delta u - \omega^2 \kappa(\vec{x})^2(\alpha - \beta i)u, \quad \alpha, \beta \in \mathbb{R}, \tag{14}$$

is used instead in the multigrid solver, and acts as a preconditioner in a suitable Krylov method. In this paper we use the pair $\alpha = 1$ and $\beta = 0.5$, which is shown in (Erlangga YA, 2006) to lead to a good compromise between approximating eq. (2) and solving the shifted system using multigrid tools. Specifically, we use a three-level geometric V-cycle and an inexact coarse-grid solution. The SL multigrid method is very consistent and robust for heterogeneous slowness models, i.e., where $\kappa$ is not uniform. However, it is considered slow and computationally expensive, especially for high wavenumbers, which is our interest.

Solving PDEs generally requires communicating information between the boundaries and the rest of the domain. However, if computation relies solely on local operations, information is limited in terms of the distance it can travel. To facilitate the transmission of information at multiple scales, multigrid methods are commonly used to solve discretized PDEs. In multigrid, solutions are defined on a hierarchy of grids, where the original fine grid $\Omega^h$ is progressively coarsened. Two distinct and complementary processes are utilized: relaxation and coarse-grid correction. Relaxation is done by performing a few iterations of a standard smoother like Jacobi or Gauss-Seidel. These smoothers have a local nature (e.g., compact convolutions), and hence are only effective at reducing part of the error. In the case of the Helmholtz system, such relaxation methods do not converge due to the indefiniteness of $A^h$, but one or two iterations of them suffice to smooth the error. The remaining components of the error typically correspond to eigenvectors of $A^h$ that are associated with small-magnitude eigenvalues, i.e., vectors $\mathbf{e}^h$ such that

$$\left\| A^h \mathbf{e}^h \right\| \ll \left\| A^h \right\| \left\| \mathbf{e}^h \right\|. \tag{15}$$

---

[1]The right-hand side $\mathbf{g}^h$ in Soubies et al. (2017) is zero.

To reduce these error components, multigrid methods use coarse-grid correction. The error $\mathbf{e}^h$ for an iterate $\mathbf{u}^h$ is estimated on a coarser grid, where it is less smooth, and interpolated back to correct $\mathbf{u}^h$ on the finer grid. In other words, we solve an instance of an error-residual equation for $\mathbf{e}^H$ projected onto the coarser grid, and then interpolate it back to the fine grid to obtain an approximate $\mathbf{e}^h$:

$$A^H \mathbf{e}^H = \mathbf{r}^H = I_h^H (\mathbf{g}^h - A^h \mathbf{u}^h), \quad \mathbf{e}^h = I_H^h \mathbf{e}^H. \tag{16}$$

The operator $A^H$ approximates $A^h$ on the coarser mesh $\Omega^H$, where $H = 2h$.

To understand why the indefiniteness of the Helmholtz problem makes multigrid inefficient, consider a smooth error on the fine grid to be a smooth eigen-mode $\mathbf{v}^h$ of $A^h$ that corresponds to a small eigenvalue $\lambda^h$. After the coarse grid correction in eq. (16), the new error is approximately (Elman et al., 2001):

$$\mathbf{e}^h = \left(1 - \frac{\lambda^h}{\lambda^H}\right) \mathbf{v}^h, \tag{17}$$

where $\lambda^H$ is the eigenvalue of $A^H$ that corresponds to the mode in $\mathbf{v}^h$ on the coarse grid. Ideally, $\lambda^H$ differs slightly from $\lambda^h$, and both are small in magnitude. However, if $\lambda^h$ and $\lambda^H$ have opposite signs, the correction is in the wrong direction and will cause the error to increase. This may happen here since $A^h$ is indefinite in our case.

To restrict a fine-grid solution to the coarse grid we use the "full-weighting" operator $I_h^H$. Conversely, to interpolate the coarse-grid solution to a finer grid, we use the bi-linear interpolation operator $I_H^h$. These operators are defined using the fixed kernels:

$$I_h^H = \frac{1}{16} \begin{bmatrix} 1 & 2 & 1 \\ 2 & 4 & 2 \\ 1 & 2 & 1 \end{bmatrix}, \quad I_H^h = \frac{1}{4} \begin{bmatrix} 1 & 2 & 1 \\ 2 & 4 & 2 \\ 1 & 2 & 1 \end{bmatrix}. \tag{18}$$

Note that these geometric operators are suitable for our problem, because the Laplacian operator in eq. (14) is homogeneous.

Taken as a whole and applied once, the above is the two-grid method, summarized in Algorithm 1. Repeated recursively, this procedure forms a cycle, termed the V-cycle. Note that relaxation is applied twice, before and after the coarse-grid correction, where it is referred to as pre- and post-relaxation, respectively. This is often done when the coarse system is still too large to solve directly. The V-cycle is often applied iteratively to solve the problem to some desired accuracy.

---

**Algorithm 1** Two-grid cycle

- Relax $v_1$ times on $A^h \mathbf{u}^h = \mathbf{g}^h$ with $\mathbf{u}^h$ as an initial guess
- $\mathbf{f}^H \leftarrow I_h^H (\mathbf{g}^h - A^h \mathbf{v}^h)$
- Solve $A^H \mathbf{e}^H = \mathbf{r}^H$ to obtain $\mathbf{e}^H$
- $\mathbf{u}^h \leftarrow \mathbf{u}^h + I_H^h \mathbf{e}^H$
- Relax $v_2$ times on $A^h \mathbf{u}^h = \mathbf{g}^h$ with $\mathbf{u}^h$ as an initial guess

---

In the classic V-cycle scheme, one may freely choose the number of levels. However, unlike other problems, the algebraically smooth error modes of the Helmholtz operator are still quite oscillatory at high wavenumbers. This means very coarse grids typically cannot represent these high-frequency error modes when about 10 grid points per wavelength are used. Thus, the performance of the solver deteriorates as the number of levels increases. For example, the results in (Calandra et al., 2013) show that three levels achieve the best balance between cost and performance.

## D  DATA GENERATION AND MULTISCALE TRAINING

Our solver network is intended to work as a preconditioner to a Krylov method, in tandem with a V-cycle. Therefore, data seen during training must be as similar as possible to the residuals seen during the Krylov method we use (FGMRES). Consider

$$\mathbf{e}^{net} = \text{SolverNet}(\mathbf{r}, \kappa^2; \theta) \tag{19}$$

to be a single forward application of the solver network for a given slowness model $\kappa(\mathbf{x})^2$ and residual $\mathbf{r}$. $\theta$ denotes the set of trainable network weights. To successfully model a good preconditioner, we seek to minimize the mean squared error (MSE)

$$\min_{\theta} \frac{1}{m} \sum_{i=1}^{m} \|\text{SolverNet}(\mathbf{r}_i, \kappa_i^2; \theta) - \mathbf{e}_i^{true}\|_2^2, \tag{20}$$

for each batch of error and residual vectors and slowness models $\{(\mathbf{e}_i^{true}, \mathbf{r}_i, \kappa_i^2)\}_{i=1}^{m}$, where $\mathbf{r}_i = A^h \mathbf{e}_i^{true}$. Since we are dealing with a linear system, where the residual can be computed simply by multiplying the error by the matrix $A^h$, the creation of the aforementioned dataset is straightforward, and there is no need to solve the PDE many times to generate ground-truth solutions. However, since the data must be as close as possible to the residuals seen during runs of the preconditioned FGMRES, we smooth the residuals in the dataset. To this end, we apply a random number of FGMRES iterations with a V-cycle preconditioner (specifically, we apply between 2 to 20 iterations). That is, starting with a random vector $\mathbf{x}_i$, we compute a RHS vector $\mathbf{b}_i = A^h \mathbf{x}_i$ and apply

$$\tilde{\mathbf{x}}_i = \text{FGMRES}(A^h, M = \text{V-cycle}, \mathbf{b}_i, \mathbf{x}^{(0)} = \mathbf{0}, iter \in \{2, ..., 20\}). \tag{21}$$

Following that, we compute error vector $\mathbf{e}_i^{true} = \mathbf{x}_i - \hat{\mathbf{x}}_i$ and residual $\mathbf{r}_i = \mathbf{b}_i - A^h \hat{\mathbf{x}}_i = A^h \mathbf{e}_i^{true}$ to obtain the $i$-th data sample. This procedure generates data samples of varying smoothness levels to be used as training residuals. Optimizing eq. (20) against these error-residual pairs directs the network to learn to handle smooth error vectors. The output of the network may be noisy (generating low errors but high residuals), hence in inference time we smooth the network's output using a V-cycle, so it is easy for FGMRES to include it when considering the optimal linear combination of the iterations.

To solve our Helmholtz problem, information must propagate from the boundaries deep into the domain, and vice-versa. This means that during training, the network must learn to propagate information across the domain regardless of the size of the domain and the location of the boundaries. Thus, networks exposed only to small domains may struggle to generalize to larger domains. While CNNs are composed of shift-invariant convolutions, there is a huge influence to the boundaries, especially in multiscale networks like U-Net that reach tiny grids. On the other hand, training on larger domains is expensive. We show that training the network on multiple scales, i.e., exposing it to varying sizes of problems, enables it to scale better to sizes unseen during training. This especially saves training time compared to training only on the maximal size since most of the iterations are obtained on smaller grid sizes. To this end, we create datasets of three different sizes: samples are taken from each source dataset, and resized to $128 \times 128$, $256 \times 256$ and $512 \times 512$ using bilinear interpolation. Samples of each respective size are considered a separate dataset for training purposes. We then alternate between these datasets during training every few epochs (here, 20). The length of the epochs containing larger training examples is adjusted to be shorter, to use mostly smaller ones, reducing the number of large examples overall. For example, an epoch of $128 \times 128$ samples is comprised of 16,000 samples, a $256 \times 256$ is made up of 10,000 samples and a $512 \times 512$ epoch is only 4000 samples long. Hence, the training requires less samples of the larger sizes while still performing well.

## D.1 SLOWNESS MODEL DATASETS

To generate slowness models for training and testing, we use three source datasets of increasing difficulty: CIFAR-10 (Krizhevsky, 2009), OpenFWI Style-A (Deng et al., 2022) and STL-10 (Coates et al., 2011). While CIFAR-10 and STL-10 are natural image datasets and therefore have no bearing on the Helmholtz problem, they serve here as large data sets of general-purpose slowness models to demonstrate our method.

Each source dataset is used to generate three target datasets of three different sizes, as mentioned. Up to 16,000 images are sampled from each dataset, resized to the appropriate size, smoothed slightly by a Gaussian kernel, and finally normalized to the range $[0.25, 1]$. Furthermore, during training only, we shift the imaginary term of $A^h$ by applying a high $\gamma$ value of 0.05. We hypothesize that training on data with a higher $\gamma$ value leads to more consistent training and yields a better convergence rate, due to the domain and boundaries being more absorbent and less reflective. Thus, the network better learns to model wave propagation and generalize to larger domains. The datasets are then split into training, validation and testing portions for use in training and inference.

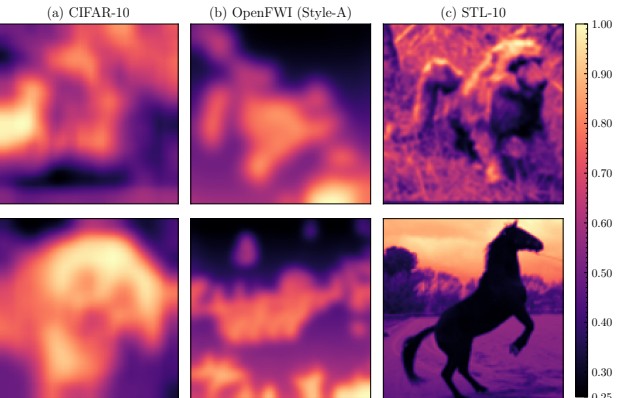

Figure 3: Example slowness models $\kappa^2$ used for training and testing: (a) models from the CIFAR-10 dataset; (b) models from the OpenFWI Style-A dataset; (c) models from the STL-10 dataset. We generate separate datasets by sampling up to 16,000 images from each of the datasets.

We scale each sample up to the appropriate size, smooth it slightly using a Gaussian kernel, and normalize the values to the range $[0.25, 1]$, as depicted in fig. 3. We note that due to the upscaling, the first two datasets (CIFAR-10 and OpenFWI Style-A) yield significantly smoother, and therefore "easier" slowness models. STL-10, while still smoothed slightly, is still quite challenging. Example slowness models derived from CIFAR-10, OpenFWI and STL-10 are shown in fig. 3.

For the out-of-distribution tests, we used the Marmousi (Brougois et al., 1990), SEG/EAGE Salt-dome, and Overthrust (Aminzadeh et al., 1997) models as out-of-distribution test problems. These models specify a spatially varying wave velocity $v$, which is inverted to give $\kappa^2 = \frac{1}{v^2}$. To match the values seen in the training of the networks, we normalized the values such that the maximum of $\kappa^2$ is 1, and took the highest frequency obeying the ten grid-point per wavelength rule.

# E   LIGHTWEIGHT ARCHITECTURE

CNN architectures in computer vision have grown over the years, with respect to both the number of parameters and FLOPs associated with the network forward application. Hence, several patterns have emerged for lighter-weight network designs. One such technique, credited to MobileNet (Howard et al., 2017), uses separable convolutions, where depthwise convolutions and channel-mixing $1 \times 1$ kernels are used separately[2]. MobileNetV2 (Sandler et al., 2018) improves upon the first version by proposing the inverted bottleneck structure, where each network "module", similar to eq. (8), consisting of three convolutions: the first is a $1 \times 1$ convolution that expands the number of channels, the second is a depthwise convolution and the last is a $1 \times 1$ convolution that shrinks the number of channels back to the previous smaller number. This sequence of operations reduces the number of parameters while still making efficient use of the hardware. We use these techniques in our solver network and in addition, we begin and end the solver network with simple downsampling and upsampling operations respectively, which are analogous to the MG prolongation and restriction operations in eq. (18). Another method we use to reduce the number of parameters is the use of a sum operation in place of a concatenation operation used in a standard U-Net where feature maps are added together. That is where feature maps computed by the encoder are used in the solver, and in the bypass connections within the solver itself, as shown in fig. 1. This use of addition instead of concatenation reduces the number of parameters in the network since the resulting number of channels remains unchanged, whereas with concatenation it is doubled.

---

[2]A depthwise convolution is a spatial convolution that is applied on each channel separately with no mixing between the channels. On the other hand, in $1 \times 1$ convolutions, there is no spatial operation, and each output channel is a simple linear combination of the input channels.

# F    ARCHITECTURE AND TRAINING DETAILS

The encoder network is essentially a multi-layer convolutional network that progressively compresses the slowness model, resulting in feature maps of sizes $16 \times \frac{I}{2} \times \frac{I}{2}$, $32 \times \frac{I}{4} \times \frac{I}{4}$ and $64 \times \frac{I}{8} \times \frac{I}{8}$, where the first number is the number of channels and $I$ is the original size of the domain along each dimension, e.g., a $2 \times 512 \times 512$ slowness model will be encoded into $16 \times 256 \times 256$, $32 \times 128 \times 128$ and $64 \times 64 \times 64$ tensors, respectively. The domains discussed here are of equal height and width for simplicity, but the design of the network does not rely on this being the case. The encoding features are computed by a learnable strided convolution followed by two more modules comprised of a learnable convolution, a batch normalization layer, and the softplus activation function. We term this module the "downsampling module".

The solver network is designed to accept these encodings as part of its architecture (see fig. 1), along with the complex-valued residual vector $\mathbf{r}$. As the computation in the solver network proceeds, the encodings are summed elementwise with the feature maps of the same dimensions. The results of the sum operation are propagated forward as feature maps.

Our solver network uses four levels, the first three of which are comprised of a learnable downsampling convolution followed by an "inverted bottleneck" module, which is discussed in appendix E. This is followed by a single downsampling module, which outputs a feature map of size $64 \times \frac{I}{16} \times \frac{I}{16}$. For smaller domains (e.g., smaller than $512 \times 512$), we skip the last downsampling operation, and set the last downsampling convolutional layer to instead maintain the size of the incoming feature map (i.e., the coarsest size is maintained at $\frac{I}{8} \times \frac{I}{8}$. We found that skipping the final downsampling operation is necessary in these cases, since otherwise the resulting feature map is too small for the implicit step to produce meaningful results, causing the solution to diverge (for example, for a $128 \times 128$ domain, the size at the coarsest level would be $8 \times 8$). Following the implicit step, the feature maps are interpolated back to the original size of the domain by three learnable upsampling modules comprised of a learnable strided transposed convolution followed by two more convolution-normalization-activation modules, with stride set to 1. Finally, the feature maps are projected back to a single complex-valued channel and upscaled once more by a non-learned bi-linear upsampling filter. The output of the solver is then the error vector $\mathbf{e}$.

We train instances of the explicit and implicit U-Nets, where each is exposed to domains of increasing size in a round-robin fashion: training alternates between $128 \times 128$, through $256 \times 256$ and finally $512 \times 512$. We train three different network instances for each dataset and network type: one is trained on $128 \times 128$ problems only, the next is trained with samples up to $256 \times 256$, and the last is trained on all three sizes. Each time, we have the same number of total samples. These tests show the influence of the training problem sizes on the test performance on larger problems, showing the importance of multiscale training. After training, each network instance, followed by a V-cycle, is tested as a preconditioner to FGMRES(10) on a batch of 1000 right-hand sides.

We train each encoder-solver pair until convergence on the validation set and up to 250 epochs. Training was done using the ADAM optimizer (Kingma & Ba, 2015) with the default parameters and batch sizes between 30–40, as GPU memory allowed. The learning rate was initialized to 0.001 and scheduled to divide by 10 every 100 epochs. For networks trained with more than one dataset (i.e., networks trained on sizes greater than $128 \times 128$), both the training and validation data was switched every 20 epochs in a round-robin fashion. We used a single current-generation consumer-grade NVIDIA GPU for each training session. All learnable tensors were initialized randomly using the default Kaiming initialization implemented by PyTorch (Paszke et al., 2019; He et al., 2015), except for the learnable kernel in the implicit layer, which was initialized to $L + iI$ where $L$ is the 5-point Laplacian kernel. This kernel was then optimized along with the rest of the network's learnable parameters. Plots of the residual MSEs (as in eq. (20)) under multiscale training are shown in fig. 4.

For the out-of-distribution tests, the networks used were trained on $512 \times 512$ domains taken from the STL-10 dataset. For this test, the sizes of the models are $512 \times 1024$ for Marmousi, $352 \times 800$ for Overthrust, and $512 \times 1024$ for the SEG/EAGE Salt-dome model. The pre-trained networks perform reasonably, but the iteration counts somewhat deteriorate compared to the in-distribution iteration counts in table 1. To improve these results, we also re-train the CNNs, i.e., optimize them for a short amount of time on the respective out-of-distribution problem. To save computation, re-training is done on models twice as small as the models used for evaluation, since the retraining is done after

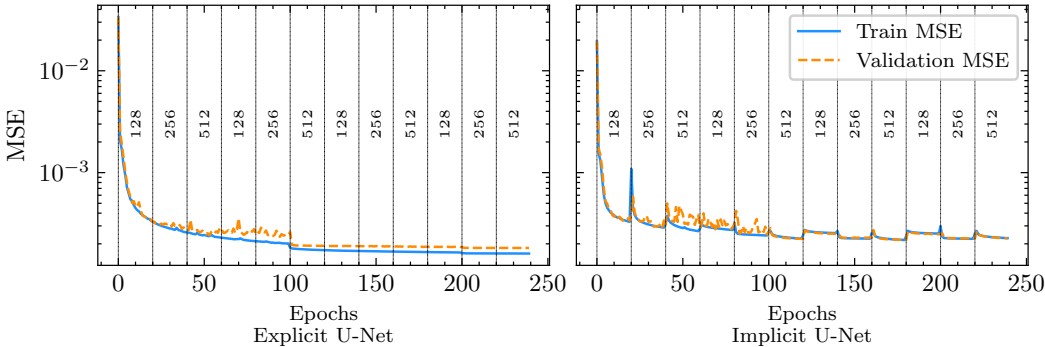

Figure 4: MSE loss during training of explicit and implicit U-Net networks with multiscale training on the OpenFWI dataset. The MSE loss is as shown in eq. (20). It is worth noting the slight increase in MSE value when the data is switched. Our results show that networks trained with multiscale training generalize better to larger unseen sizes.

the model is known, at solve time. To this end, we generate 300 pairs of error vectors and their corresponding residuals, and re-train the model for 30 epochs, each comprised of these 300 vector pairs. Then, we evaluate the networks on new error-residual pairs on the original larger-sized test cases. Figure 2 shows how performance on these problems is improved significantly compared to the original networks, at the small cost of additional training. Here, as well, the implicit network has an advantage over the explicit one.

## G    ADDITIONAL RESULTS

### G.1    WALL-CLOCK RUNTIME COMPARISON PER ITERATION ON A GPU

In our last experiment, we compare the runtime performance of our networks and the other methods considered in this paper. To this end, the various methods are run as a preconditioner during FGMRES. The wall-clock time to solution is measured and divided by the number of iterations performed. We report the average time of a preconditioned FGMRES(10) outer iteration, which includes ten preconditioning steps. The experiment is performed 100 times and the results are averaged. Table 2 lists the average runtimes per iteration as well as other statistics such as the number of parameters and FLOPs in the network. The timings do not include the application of the encoder network, which is applied only once per given linear discrete operator (defined by a slowness model $\kappa(\mathbf{x})$), and its output serves all iterations and right-hand-sides with that operator. Hence, its inference time is insignificant compared to the total solution time. Fore completeness, we report the measures of the encoder network in Table 3. Note that while the implicit U-Net appears to be more expensive than the explicit U-Net, it is likely that a custom implementation, e.g., (Treister et al., 2018), of the implicit layer will eliminate much of this performance gap, as the lone implicit layer on the U-Net's coarsest grid is obtained on small feature maps. The results presented here are given for a standard GPU-based PyTorch implementation.

Table 2: Runtime comparison of solution methods. The runtime per outer iteration of FGMRES(10) is averaged over 100 right-hand sides and 100 iterations per RHS. FGMRES was run for a random slowness model and for grid sizes up to $4096 \times 4096$ with the preconditioner listed by each respective line. Where neural networks are used as preconditioner, they are also followed by an augmenting application of a V-cycle. All standard deviations are too insignificant to list.

| Preconditioner | Params. | FLOPs | Runtime avg. (s) | | | |
|---|---|---|---|---|---|---|
| Test grid size | | | $512^2$ | $1K^2$ | $2K^2$ | $4K^2$ |
| V-cycle only | N/A | 46M | .06 | .08 | .13 | .33 |
| Azulay & Treister (2021) | 2.5M | 3.14B | .08 | .12 | .29 | .98 |
| Explicit model | 360K | 245.8M | .07 | .09 | .19 | .57 |
| Implicit model | 360K | 245.8M | .07 | .09 | .19 | .59 |

Table 3: Runtime of encoding slowness models. The runtime cost in seconds of applying the encoder to a slowness model of various grid sizes. The cost measurements are averaged over 100 random slowness models. All standard deviations are too insignificant to list.

| Encoder | Params. | FLOPs | Runtime avg. (s) | | | |
|---|---|---|---|---|---|---|
| Test grid size | | | $512^2$ | $1K^2$ | $2K^2$ | $4K^2$ |
| Azulay & Treister (2021) | 1.9M | 5.9B | .003 | .01 | .041 | .17 |
| Ours | 1.2M | 504M | .001 | .003 | .013 | .052 |

