# OpenReview forum: "Multigrid-Augmented Deep Learning Preconditioners for the Helmholtz Equation using Compact Implicit Layers"
_ICLR.cc/2024/Workshop/AI4DiffEqtnsInSci — AI4DiffEqtnsInSci @ ICLR 2024 Poster_

### Official Review · Reviewer_RRqp · 2024-02-28
**Solid paper, no concerns, recommend acceptance**

**Rating:** 7
**Confidence:** 3

**Review:**

**Explanation of paper**

This paper uses a specially designed neural network as a preconditioner for the Helmholtz equation, which a PDE typically solved via iterative methods. This paper is similar to Azulay and Treister (2021), the main difference being an implicit layer is added to the U-Net preconditioner, which improves performance.

**Pros**

* When I review papers for these workshops, more often than not I find serious flaws or methodological issues with the paper. In this paper, I see no major flaws and have no concerns.
* This is a good, solid paper, that tackles a problem of relevance to this workshop while comparing to a strong baseline and using good empirical practices.
* While the Helmholtz equation is not commonly studied by the AI4DiffEq research community, it is an important PDE of interest to many disciplines of engineering and science. Furthermore, the strategy here of using a neural network as a preconditioner *is* very common in the AI4DiffEq community; those papers typically consider the (simpler) Poisson equation.
* These authors appear to be experts in preconditioners and iterative numerical methods for elliptic PDEs. Their presence at this workshop will likely add value to the community.
* I think the introduction is excellent, and is helpful both for explaining the context this work is in as well as for future authors working on the Helmholtz equation.

**Cons**

* I found section 3 of this paper hard to understand. Maybe this is my fault -- I was an emergency reviewer for this paper and didn't have enough time to understand the architectural details -- but I do think that these ideas could be explained with improved clarity.
* If this were the main ICLR conference, I would feel the novelty of this paper (i.e., mainly an architectural modification) is not sufficient to justify acceptance; in other words, this paper is to some extent salami slicing. However, for a workshop, I think this is an appropriate contribution.

**Additional comments**
* "Preconditionding" in title is spelled incorrectly. Even if it were spelled correctly, the PDF uses the word "preconditioners" not "preconditioning".

**Conclusion**

My recommendation is to accept this paper. It is a solid paper, which uses good empirical practices, a strong baseline, is clearly within the scope of the workshop, and has nice experiments and good results. If I were argue against acceptance, I would say that the results in this paper are not of enormous significance and I have some concern about salami-slicing. However, a vast majority of papers (98%+) are not of great significance, so I think that is perfectly okay for a workshop.

---

### Meta-Review · Area_Chair_2Wac · 2024-03-03

**Recommendation:** Accept (Poster)

**Metareview:**

The reviewer marks this paper as a clear acceptance and I agree. I encourage the authors to address the reviewer's comments in the camera-ready version.

---

### Decision · Program_Chairs · 2024-03-03

Accept (Poster)